# βIII-Tubulin Structural Domains Regulate Mitochondrial Network Architecture in an Isotype-Specific Manner

**DOI:** 10.3390/cells11050776

**Published:** 2022-02-23

**Authors:** Amelia L. Parker, Wee Siang Teo, Simon Brayford, Ullhas K. Moorthi, Senthil Arumugam, Charles Ferguson, Robert G. Parton, Joshua A. McCarroll, Maria Kavallaris

**Affiliations:** 1Children’s Cancer Institute, Lowy Cancer Research Centre, UNSW Sydney, Sydney, NSW 2052, Australia; am.parker@garvan.org.au (A.L.P.); wee.teo@glytherix.com (W.S.T.); sbrayford@ccia.org.au (S.B.); 2School of Women’s and Children’s Health, Faculty of Medicine and Health, UNSW Sydney, Sydney, NSW 2052, Australia; 3Australian Centre for NanoMedicine, UNSW Sydney, Sydney, NSW 2052, Australia; 4Monash Biomedicine Discovery Institute, Faculty of Medicine, Nursing and Health Sciences, Monash University, Clayton, Melbourne, VIC 3800, Australia; ullhas.moorthi@monash.edu (U.K.M.); senthil.arumugam@monash.edu (S.A.); 5European Molecular Biological Laboratory Australia (EMBL Australia), Monash University, Clayton, Melbourne, VIC 3800, Australia; 6Institute for Molecular Bioscience, University of Queensland, Brisbane, QLD 4072, Australia; c.ferguson@imb.uq.edu.au (C.F.); r.parton@imb.uq.edu.au (R.G.P.); 7Centre for Microscopy and Microanalysis, University of Queensland, Brisbane, QLD 4072, Australia

**Keywords:** tubulin isotype, mitochondria, microtubules, carboxy-terminal tail

## Abstract

βIII-tubulin is a neuronal microtubule protein that is aberrantly expressed in epithelial cancers. The microtubule network is implicated in regulating the architecture and dynamics of the mitochondrial network, although the isotype-specific role for β-tubulin proteins that constitute this microtubule network remains unclear. High-resolution electron microscopy revealed that manipulation of βIII-tubulin expression levels impacts the volume and shape of mitochondria. Analysis of the structural domains of the protein identifies that the C-terminal tail of βIII-tubulin, which distinguishes this protein from other β-tubulin isotypes, significantly contributes to the isotype-specific effects of βIII-tubulin on mitochondrial architecture. Mass spectrometry analysis of protein–protein interactions with β-tubulin isotypes identifies that βIII-tubulin specifically interacts with regulators of mitochondrial dynamics that may mediate these functional effects. Advanced quantitative dynamic lattice light sheet imaging of the mitochondrial network reveals that βIII-tubulin promotes a more dynamic and extended reticular mitochondrial network, and regulates mitochondrial volume. A regulatory role for the βIII-tubulin C-terminal tail in mitochondrial network dynamics and architecture has widespread implications for the maintenance of mitochondrial homeostasis in health and disease.

## 1. Introduction

Microtubules are fundamental components of all eukaryotic cells. In humans, microtubules are composed of mixed combinations of eight α-tubulin isotypes and seven β-tubulin isotypes [1]. βIII-tubulin has a highly specific tissue distribution, and is expressed in neurons and testicular Sertoli cells in adults. It is also aberrantly expressed in epithelial cancers derived from tissues that normally lack or have very low expression of βIII-tubulin, including non-small cell lung cancer [2]. Despite recent advances identifying that β-tubulin isotypes have specific functions as regulators of microtubule dynamics, metabolism and chemosensitivity [3,4,5,6], the distinct isotype-specific roles of β-tubulin isotypes remain incompletely characterized.

The architecture and dynamics of the mitochondrial network have far-reaching impacts on a broad range of fundamental cellular functions in health and disease. Microtubules are involved in mitochondrial localization and trafficking [7,8,9,10], and tubulin proteins have also been associated with the mitochondrial quality control machinery [11,12,13,14], although the contribution of individual β-tubulin isotypes on mitochondrial dynamics remains unclear. Mitochondrial trafficking along microtubules is highly regulated to direct mitochondria to sites of high metabolic demand [9,15,16,17]. This interaction between microtubules and mitochondria is a reciprocal relationship, with microtubules involved in mitochondrial trafficking and degradation, and these processes influencing microtubule stability and tubulin degradation in return [12]. The role of specific tubulin isotypes in regulating mitochondrial trafficking on microtubules is poorly understood.

The β-tubulin isotypes that form microtubules share a high degree of sequence similarity, and are distinguished from one another largely by the sequence of their C-terminal tails that extend outwards from the wall of the microtubule. Protein–protein interactions with these isotype-specific sequences are thought to confer diverse and unique functionality to the different β-tubulin isotypes and explain their tissue specific distributions [6]. Growing associative evidence points to an isotype-specific role for β-tubulin proteins in mitochondrial trafficking and function, where the composition of tubulin isotypes within cells collectively and combinatorially regulates microtubule-dependent functions [18,19]. Recent studies have indicated that the posttranslational modifications on microtubule subsets can regulate ER localization as well as that of lysosomes and vesicles through specific protein–protein interactions [20]. In addition to post-translational modifications, different β-tubulin isotypes also confer heterogeneity to the microtubule network to collectively constitute the tubulin code. The β-tubulin isotypes differentially affect microtubule dynamics [3,21,22,23], which regulates kinesin-independent mitochondrial trafficking [24,25,26] by hitherto undefined mechanisms.

Similarly, both the β-tubulin body and tail regions interact with kinesin proteins to regulate kinesin processivity [27,28,29,30], thereby affecting kinesin-dependent trafficking [31,32]. β-tubulin proteins also interact with the mitochondrially localized voltage-dependent anion channel to regulate channel conductance in an isotype-specific manner [33,34,35]. In these respects, the βIII-tubulin isotype displays distinct characteristics compared with other β-tubulin isotypes. βIII-tubulin, and particularly its C-terminal tail domain, promotes more dynamic microtubules [3,23], and orchestrates the collective behavior of the microtubule network [3], significantly blocks the mitochondrial voltage-dependent anion channel [33], and has a greater inhibitory effect on kinesin-1 processivity than other β-tubulin isotypes [30]. βIII-tubulin is also enriched in mitochondrial membranes [7,34,36,37,38], although it is not clear if βIII-tubulin interacts directly with mitochondrial membranes or through linker proteins [7,9,34,36,39]. These functional effects of βIII-tubulin on the regulators of mitochondrial trafficking are re-enforced by clinical observations that congenital mutations within helix 12 adjacent to the βIII-tubulin C-terminal tail result in neurodevelopmental defects associated with abnormal mitochondrial trafficking, and phenocopy those caused by kinesin mutations [40,41,42,43]. However, the role of βIII-tubulin structural domains in the wild-type protein remains unclear. 

Our previous work identified a role for βIII-tubulin in promoting central carbon metabolism flexibility by enabling cancer cells to rapidly switch between oxidative phosphorylation and glycolysis in response to nutrient starvation independently of microtubule architecture [4]. This warrants further investigation of how this β-tubulin isotype affects mitochondrial architecture and, therefore, function. However, the role of βIII-tubulin in mitochondrial dynamics, and the importance of its structural domains in regulating mitochondrial network architecture, remain unclear. 

This study used orthogonal high-resolution static and dynamic imaging approaches to identify an isotype-specific role for βIII-tubulin and its structural domains in regulating mitochondrial localization and architecture, thereby contributing to our understanding of the tubulin code. 

## 2. Materials and Methods

### 2.1. Cell Culture

The human non-small cell lung cancer (NSCLC) cell lines NCI-H460 and A549 cells were cultured and validated as described previously [4,5]. All parental, control and knockdown cell lines were validated by STR profiling (ATCC). Control non-silencing shRNA and βIII-tubulin shRNA expressing NCI-H460 and A549 clones were cultured and validated as described previously [4,5]. βIII-tubulin is a minor β-tubulin isotype, and we have previously validated that suppression of βIII-tubulin expression in these cells does not alter the expression of other β-tubulin isotypes [5,44]. Gene-edited NCI-H460 cell lines were generated and cultured as described previously [3]. Two sets of gene-edited NCI-H460 cells were generated, each set containing clones with matched expression of the modified βIII-tubulin from the *TUBB3* locus as previously described [3]. Importantly, the expression of other β-tubulin isotypes was not altered by the expression of modified βIII-tubulin proteins from the edited *TUBB3* gene [3]. Gene editing of the *TUBB3* locus in the gene-edited clones was confirmed by genomic sequencing (Sanger sequencing, Garvan Molecular Genetics, Garvan Institute, Australia) across the edited site. All cell lines were regularly screened to ensure the absence of mycoplasma contamination using the MycoAlert MycoPlasma Detection Kit (Lonza, Switzerland).

### 2.2. Mitochondrial Network Immunofluorescence

NCI-H460 and A549 cells cultured on poly-D-lysine-coated chamber slides (ThermoFisher Scientific, Waltham, MA, USA) were fixed in 4% paraformaldehyde and permeabilized in 0.1% Triton X-100, blocked in 10% FCS/PBS at room temperature followed by immunostaining with GRP75 antibody (JG1, Abcam, Cambridge, UK; 1:500) overnight at 4°C. Cells were washed three times with PBS and incubated with Alexa-fluor568-conjugated secondary antibody (1:500) for one hour at room temperature. Slides were then washed (PBS) and mounted in DAPI-containing mounting media (Vectashield, Vector Labs, Burlingame, CA, USA). Cells were imaged using a Leica SP8 confocal microscope.

For quantification of mitochondrial distribution across the entire cytoplasm, the GRP75 staining intensity was measured radially for each cell as described previously [20]. All images were analyzed in a blinded fashion. Briefly, a FIJI macro was used to define the center of the nucleus and remove the signal of neighboring cells. From the manually defined center point, the GRP75 signal intensity was mapped to polar co-ordinates with re-scaling to correct for artifacts generated by the square pixel shape. The location of the edge of the cell and edge of the nuclear envelope were defined in polar co-ordinates. The fluorescence signal was then re-scaled to a normalized axis where the center of the cell to the cell periphery corresponded to 0 to 100%. In addition, the nucleoplasm was scaled as 0 to 25%, as a control for nuclear size. The mean distribution radius was calculated using Matlab scripts as previously described [20]. The morphology of the mitochondrial network as reticular, rounded, or both reticular and rounded (i.e. mixed morphology) was manually scored in a blinded fashion. 

### 2.3. Transmission Electron Microscopy

The preparation of cells for transmission electron microscopy was performed as described previously [45]. Briefly, NCI-H460 and A549 cells were plated onto tissue culture-treated plastic. Seventy-two hours later, cells were rinsed quickly in warm Sorensen’s Buffer and fixed in pre-warmed (37 °C) 2.5% glutaraldehyde/ 2% paraformaldehyde solution (Proscitech, Kirwan, Australia) in Sorensen’s Buffer for 1 h at room temperature. Cells were further processed as described previously [46]. 60 nm ultrathin sections were cut on an Ultracut 6 (Leica Microsystems, Singapore) ultramicrotome. Grids were imaged at 80 kV on a JEOL 1011 transmission electron microscope fitted with a Morada 4 K × 4 K Soft Imaging Camera at two-fold binning (Olympus, Hong Kong). Mitochondrial volume density as a percentage of cytoplasmic volume was determined by stereology using point counting with a double lattice grid on systematically captured random images. Percentage mitochondrial area was averaged for 8–23 cells per biological replicate for *n* = 3 (A549) and *n* = 2 (NCI-H460) biological replicates (separate cell cultures).

### 2.4. Preparation of NCI-H460 cells for Lattice Light-Sheet Imaging of Mitochondria

NCI-H460 cells (NCI-H460 Ctrl_SH2_ or NCI-H460 βIII_SH4_) were seeded in 6-well plates at 2.5 × 10^5^ cells per well and incubated 24 h to allow attachment. For characterizing overall mitochondrial numbers and volumes, cells were incubated with MitoTracker Green (M7514, Invitrogen) according to the manufacturer’s instructions. For assaying fusion between mitochondria, cells were transfected with the photo-activatable mitochondrial marker, mito-Dendra2 (Addgene #55796) at 1 μg DNA per well using Lipofectamine 2000 (Thermo Fisher, Waltham, MA, USA) according to the manufacturer’s instructions. Following 24 h incubation, cells were detached via trypsinization, counted, and seeded again into 6-well plates containing several 5 mm glass coverslips at 1 × 10^5^ cells per well, and allowed a further 24 h to attach to the coverslips before commencing lattice light-sheet imaging.

### 2.5. Lattice-Light Sheet Imaging Mitochondrial Dynamics 

Cells were imaged using a lattice light sheet microscope (3i, Denver, CO). Excitation for MitoTracker Green or Dendra-2 was achieved using 488 nm diode lasers (MPB communications) at 1–5% AOTF transmittance through an excitation objective (Special Optics 28.6 × 0.7 NA 3.74-mm immersion lens) and detected via a Nikon CFI Apo LWD 25 × 1.1 NA water immersion lens with a 2.5 × tube lens. Live cells were imaged in 9 mL of 37 °C-heated DMEM and were acquired with 2 × Hamamatsu Orca Flash 4.0 V2 sCMOS cameras with a dichroic at 560 nm. Photoconversion of Dendra-2 was achieved by interleaving the excitation sequence with 488 nm Beam at high laser power in single Bessel beam mode. 

### 2.6. Analysis of Lattice-Light Sheet Imaging of Mitochondrial Dynamics

Segmentation and tracking of mitochondria were performed using a combination of ORS Dragonfly and custom codes. Image threshold was performed at upper Otsu and eroded in 3D with a kernel size of 3. Speckle noise and backgrounds over thresholds were removed by removing voxel counts less than 10. Tracking was performed by custom MATLAB codes based on CME analysis [47]. Statistical analysis comparing control cells with those expressing βIII-tubulin-targeted shRNA was performed using two-way ANOVA or Mann–Whitney U test for multiple and single comparisons, respectively. 

### 2.7. Code Availability

Information and requests for Matlab codes should be directed to the Lead Contact Maria Kavallaris.

### 2.8. Statistical Analysis

Data are presented as the mean ± standard error of the mean (SEM) unless otherwise stated. Data were analyzed using a Kruskal–Wallis test (non-parametric ANOVA) with Dunn’s multiple comparison test for more than two groups or Student’s *t*-test for two group comparisons where appropriate (GraphPad Prism 5, Graphpad Software Inc., La Jolla, CA, USA). Analysis of mass spectrometry data was performed in the software described above, and in R (v3.6.3). A *p*-value less than 0.05 was considered statistically significant. The Bonferroni–Hochberg method was used to correct *p*-values for multiple comparisons. 

Further information and requests for resources and reagents should be directed to, and will be fulfilled by, the Lead Contact, Maria Kavallaris.

## 3. Results

### 3.1. βIII-Tubulin Regulates the Mitochondrial Network Structure and Morphology

The role of βIII-tubulin in regulating the mitochondrial network was investigated in two independent non-small cell lung cancer (NSCLC) cell lines with endogenous expression of βIII-tubulin using stable specific knockdown of this isotype [4]. As indicated in previous studies, stable βIII-tubulin suppression of approximately 90% was achieved in these isogenic A549 and H460 non-small cell lung cancer cell lines [4], with no perturbation in the expression of other β-tubulin isotypes [23], microtubule cytoskeletal architecture [48], microtubule dynamics or proliferation rates in steady-state normal growth conditions [5,23,44]. However, the effect of βIII-tubulin on mitochondrial architecture had not yet been explored. To visualize the mitochondrial network, immunofluorescence staining for the mitochondrial marker mtHSP70/GRP75 [49,50] was performed. Comparison of control (H460 Ctrl_SH2_; A549 Ctrl_SH27_) cells and cells with suppressed levels of βIII-tubulin (H460 βIII_SH4_; A549 βIII_SH61_) (Figure 1A, Appendix A) identified that mitochondria form a tubular, reticular network that extends to the cell periphery in NCI-H460 and A549 control cells, with the A549 cell line in general having a more centralized mitochondrial network compared with the NCI-H460 cell line (Figure 1A). Comparatively, the mitochondrial network appeared subtly but nonetheless discernibly more perinuclear in NSCLC cells with suppressed βIII-tubulin expression compared with control cells (Figure 1A). 

To generate a quantitative, robust and unbiased characterization of the mitochondrial distribution throughout the cytoplasm, we applied a recently developed image analysis algorithm to immunofluorescence images of control and βIII-tubulin knock-down cells stained with the mitochondrial marker GRP75 to quantify the mitochondrial distribution radially across the entirety of each cell [20] (Figure 1B,Ci,Cii). The mean distribution radius parameter calculated from this analysis provides a measure of mitochondrial distribution towards the cell periphery, and a larger radius indicates that mitochondria are localized towards the cell periphery (Figure 1Biv). This analysis indicated that the mitochondrial network of cells with suppressed βIII-tubulin expression was more perinuclear than control cells (Figure 1D).

Analysis of the morphology of this mitochondrial network indicated that the network was also more fragmented and rounded in cells with suppressed levels of βIII-tubulin compared to the more reticular mitochondrial network structure in control cells (Figure 1E).

This phenotype, consisting of a more perinuclear and rounded mitochondrial network in βIII-tubulin knockdown cells, was exacerbated in glucose starvation conditions in H460 cells (Appendix A), with glucose starvation known to induce mitochondrial network reorganization to respond to the bioenergetic and endoplasmic reticulum stress induced by nutrient deprivation. Mitochondrial rounding, network fragmentation and a centralized mitochondrial distribution in βIII-tubulin knockdown cells is consistent with mitochondrial membrane potential depolarization and retrograde transport of mitochondria targeted for degradation [16,51,52,53,54]. However, quantitation of mitochondrial membrane potential using JC-1 staining and FACS analysis showed that βIII-tubulin expression did not affect the mitochondrial membrane potential (Appendix A). Together, this indicates that βIII-tubulin expression enables a more extended reticular mitochondrial network in NSCLC cells. 

### 3.2. βIII-Tubulin Regulates Mitochondrial Volume 

To examine the effect of βIII-tubulin expression on mitochondrial ultrastructure and volume in more detail, we performed transmission electron microscopy (TEM) on cells with endogenous and suppressed expression of βIII-tubulin. Consistent with the light microscopy analysis (Figure 1), mitochondria appeared more perinuclear in A549 βIII-tubulin knockdown cells (Figure 2A and Appendix A). Similarly, there was a trend towards mitochondria being less tubular and more rounded in knockdown cells compared with control cells (Figure 2B, *p* = 0.12). In addition, precise quantitation of mitochondrial volume showed that suppression of βIII-tubulin expression significantly reduced the mitochondrial volume in A549 (50.69 ± 12.29% mean ± SD, *p* = 0.003) (Figure 2A,Ci) and NCI-H460 cells (70.07± 10.26%, *p* = 0.0459) (Figure 2Cii) compared with control cells. 

### 3.3. The βIII-Tubulin C-Terminal Tail Regulates the Mitochondrial Network Structure and Morphology

The C-terminal tail of the βIII-tubulin protein distinguishes this tubulin isotype from other β-tubulin isotypes [3]. In order to examine the differential importance of the βIII-tubulin body and C-terminal tail structural domains on the mitochondrial network, gene-edited NCI-H460 cell lines were engineered to express the full-length βIII-tubulin protein (ZB3) compared with cell lines gene-edited to express a truncated version of the protein lacking the C-terminal tail region from Ala429 (ZB3Δ), or with the βI-tubulin C-terminal tail sequence in substitution for the βIII-tubulin sequence (ZB3/CB1) or with the βIII-tubulin body replaced with the βI-tubulin body (ZB1/CB3) [3]. These genetically modified cell lines were grouped into two panels with matched expression levels, with each panel expressing the modified proteins at 30% and 100% of the endogenous βIII-tubulin expression, as described previously [3]. Replacement of the endogenous βIII-tubulin gene expression using gene-editing did not affect cell proliferation, microtubule cytoskeletal architecture or dynamics relative to the parental NCI-H460 cell line, confirming that the gene-editing approach itself does not artefactually affect microtubule behavior or cell proliferation [3]. Modification of the βIII-tubulin structural domains also did not affect the β-tubulin isotype composition or actin cytoskeletal architecture in normal growth conditions (data not shown).

To examine if the structural domains of βIII-tubulin affect the distribution of mitochondria in the cell, we applied the quantitative, whole-cell analysis method described above to immunofluorescence images of these gene-edited cells stained with the mitochondrial marker GRP75. It identified that, similarly to our findings in cells with suppressed expression of the full-length βIII-tubulin protein (Figure 1 and Figure 2), loss of the βIII-tubulin C-terminal tail either through truncation of the protein (ZB3Δ) or replacement with the βI-tubulin C-terminal tail (ZB3/CB1) resulted in subtle but significant localization of the mitochondrial network to the perinuclear region compared with the full-length protein (ZB3; Figure 3A–C). Conversely, replacement of the βIII-tubulin body with the βI-tubulin sequence (ZB1/CB3) partially extended the mitochondrial network towards the cell periphery in one expression-matched set of clones (Figure 3Bi), further supporting a role for the βIII-tubulin tail itself in promoting an extended mitochondrial network.

Morphologically, loss of the βIII-tubulin C-terminal tail, either by truncation of the tail (ZB3Δ) or replacement of the C-terminal tail with the βI-tubulin sequence (ZB3/CB1), promoted a more fragmented and rounded mitochondrial network morphology (Figure 3C) consistent with our findings in cells with suppressed expression of βIII-tubulin. Replacement of the βIII-tubulin body with the βI-tubulin body also produced a more rounded and fragmented mitochondrial network compared with cells expressing the full-length βIII-tubulin protein (Figure 3C), but its effect was not as pronounced as that induced by the C-terminal tail sequence. This indicates that while the βIII-tubulin body and C-terminal tail both contribute to a more extended reticular mitochondrial network, the sequence of the βIII-tubulin C-terminal tail, not just its presence or absence, more significantly regulates mitochondrial network localization and architecture than the tubulin body in an isotype-specific manner. 

The C-terminal tail of β-tubulin proteins is a site for protein–protein interactions that are thought to confer as yet poorly defined isotype-specific functionality [6]. To explore protein–protein interactions that occur specifically with the βIII-tubulin isotype, we performed β-tubulin immunoprecipitation and proteomic mass spectrometry analysis (Figure 3F). As expected, this analysis identified α-, β-- and γ-tubulin isotypes expressed in this cell line and known to be associated with βIII-tubulin and βI-tubulin (TBA, TBA1C, TBA3C, TBA3E, TBA8, TBB3, TBB5, TBB, TBB1, TBB6, TBB4B, TBB2, TBG1, TBG2) as well as microtubule associated proteins (MAP4, MAP1A) as pulled down with both βI-tubulin and βIII-tubulin proteins. This confirmed enrichment of the target β-tubulin isotype of interest while retaining interactions between these tubulin isotypes and microtubule-interacting proteins.

Importantly, we identified 14 proteins that specifically interact with βIII-tubulin but not with βI-tubulin (Appendix A, Appendix A). Functional annotation analysis identified that these βIII-tubulin-associated proteins were significantly enriched in the mitochondrial matrix and the perinuclear region of the cytoplasm (Appendix A), consistent with a role for βIII-tubulin in regulating mitochondrial network architecture and trafficking of mitochondria away from perinuclear locations. While the majority of these proteins are known to be localized to the outer mitochondrial membrane and cytoplasm and therefore have the potential to interact directly with βIII-tubulin, HAGH is localized to the mitochondrial matrix, so its interactions may be facilitated by other, as yet unidentified, scaffolding proteins (Appendix A). Interestingly, this analysis identified a regulator of mitochondrial dynamics, PARK7/DJ-1, as associated with βIII-tubulin but not βI-tubulin (Figure 3F,H, Appendix A). The cytoplasmic localization of this protein [55,56], means it has the potential to interact with βIII-tubulin. Protein interaction network analysis identified this protein as central to a broader network of proteins that also specifically interact with βIII-tubulin (Figure 3F), suggesting that protein–protein interactions specific to βIII-tubulin have the potential to directly affect mitochondrial network architecture. Further validation of this finding may confirm if a specific interaction of this mitochondrial regulator with this tubulin isotype contributes to the regulation of mitochondrial architecture by βIII-tubulin. Other proteins in this interaction network may also indirectly modulate mitochondrial volume and architecture through βIII-tubulin. The actin cytoskeletal regulator tropomyosin 3A was also pulled down with βIII-tubulin (Figure 3F). The actin cytoskeleton is known to regulate mitochondrial dynamics and, therefore, altered actin dynamics may indirectly mediate the effects of βIII-tubulin. These findings warrant further investigation into the direct and indirect molecular mechanisms that may contribute to βIII-tubulin-mediated mitochondrial regulation.

### 3.4. βIII-Tubulin Promotes a Dynamic Mitochondrial Network

Mitochondria localization and network architecture result from changes in the fission, fusion and trafficking of mitochondria throughout the cytoplasm, enabling adaptive responses to stimuli and cellular stress. To determine if these effects of βIII-tubulin on the mitochondrial network architecture and volume may reflect its effects on mitochondrial network dynamics, we used dynamic lattice light sheet microscopy (LLSM) to quantitatively compare mitochondrial dynamics in real time in cells with suppressed βIII-tubulin expression compared with control cells. LLSM allows for fast, volumetric imaging necessary to capture the total number of mitochondria and their dynamic behavior throughout the entire volume of each cell (Figure 4A–C). By segmenting the intensity of MitoTracker Green (Figure 4B), we quantified the volume of individual mitochondria as well as the total number of mitochondria in the whole volume of the cell for control and βIII-tubulin knockdown cells (Figure 4C). Consistent with our previous observations in static immunofluorescence analysis (Figure 1), this dynamic imaging modality also revealed a fragmented mitochondrial network in cells with suppressed βIII-tubulin expression (Figure 4C, Supplementary movies). The mean volume of mitochondria in βIII-tubulin knockdown cells was also significantly reduced (1.9 ± 1.3 µm^3^) compared with control cells (3.8 ± 3.1 µm^3^, *p* < 0.01, Figure 4D), as measured by TEM (Figure 2). Similarly, the number of mitochondria in control cells was approximately half of that in βIII-tubulin knockdown cells at any single time point (Figure 4E), together reflecting a more fragmented mitochondrial network in βIII-tubulin knockdown cells (Figure 4C). 

Perturbations in mitochondrial velocity, fission and fusion dynamics can result in the strong effects on the size and number of mitochondria that we observed in βIII-tubulin knockdown cells. Therefore, we further investigated if βIII-tubulin expression affected mitochondrial motility and dynamics. We utilized the photoconvertible protein Mito-Dendra2 to precisely track the motility and fusion dynamics of individual mitochondria within the cytoplasm of control and βIII-tubulin knockdown cells (Figure 4F). The use of Mito-Dendra2 provides superior quantitation of mitochondrial dynamics compared with the use of MitoTracker Green (Figure 4A,B) by focusing on photoconverted mitochondria within a volume containing dense, dynamic non-photoconverted mitochondria (Figure 4F–H). There was a consistent trend towards reduced maximal mitochondrial velocities in βIII-tubulin knockdown cells (0.37 µm/s) compared with control cells (0.51 µm/s) (*p* = 0.0612, Figure 4I). By segmenting out the signal of photoconverted Dendra2 channel and measuring the total intensity of individual mitochondria in this channel as well as the non-photoconverted channel, we counted the number of double-colored mitochondria (Figure 4G,H). Since βIII-tubulin knockdown cells have double the number of mitochondria compared with control cells (Figure 4E), it would be expected that if the rate of mitochondrial fission and fusion are equivalent in the two cell lines, then the growth in the number of double-colored mitochondria in βIII-tubulin knockdown cells should be approximately double that of control cells. However, we observed that the growth of double-labelled mitochondria occurred at only a marginally higher rate in βIII-tubulin knockdown cells compared with control cells (Figure 4J). Therefore, we reasoned that mitochondrial fusion still occurs when βIII-tubulin expression is suppressed and the potential effect of βIII-tubulin expression in increasing mitochondrial fusion dynamics requires a more sensitive approach to discern more subtle effects.

Overall, the quantitative imaging modalities employed in this work reveal an isotype-specific role for the βIII-tubulin protein, mediated principally by its C-terminal tail region, in regulating mitochondrial number, size and speed. Protein–protein interaction analysis indicates that this tubulin isotype specifically interacts with regulators of mitochondrial dynamics. Together this provides new insight into an isotype-specific role for βIII-tubulin in regulating the dynamics of the mitochondrial network by advancing our understanding of the tubulin code.

## 4. Discussion

The microtubule network integrates and regulates diverse intracellular functions at all stages of the cell cycle, yet the role of individual β-tubulin isotypes in health and disease remains unclear. In particular, the role of β-tubulin isotypes and their structural domains on mitochondrial architecture and dynamics in unknown. This study has identified a specific role for βIII-tubulin and its C-terminal tail in regulating mitochondrial dynamics, and promoting an extended, interconnected, and dynamic reticular mitochondrial network architecture. This is the first demonstration, to our knowledge, that the βIII-tubulin protein, and in particular its C-terminal tail, affects mitochondrial volume and architecture within the complex intracellular environment. 

Comparative analysis of βIII-tubulin with βI-tubulin highlights that the sequence of the C-terminal tail of βIII-tubulin plays a more profound regulatory role than the tubulin body in regulating mitochondrial dynamics, and is consistent with this region of the protein being highly divergent compared with other β-tubulin isotypes. These findings are consistent with studies in reduced systems that have identified that the βIII-tubulin C-terminal tail reduces kinesin-1 processivity in an isotype specific manner [30], with kinesin proteins acting as important mediators of mitochondrial trafficking [31,32]. Our observations that both the βIII-tubulin body and C-terminal tail regulate mitochondrial dynamics may reflect the importance of both these β-tubulin regions as interacting sites for kinesin motors [18,19]. However, the more pronounced effect of the C-terminal tail on mitochondrial dynamics compared with the βIII-tubulin body is supported by observations that mutation of the βIII-tubulin helix 12 adjacent to the C-terminal tail region decreases mitochondrial trafficking to the axonal growth cone due to altered kinesin interactions in neurons [43]. Whether βIII-tubulin promotes a more extended and interconnected mitochondrial architecture through kinesin-dependent or -independent mechanisms remains an active area of research in the laboratory. We have previously demonstrated that the βIII-tubulin C-terminal tail regulates microtubule dynamics [3] and this indirect, kinesin-independent mechanism may contribute to the effects of this tubulin isotype on mitochondrial dynamics. In addition, the tethering of mitochondria to microtubules by as yet undefined mechanisms regulates both mitochondrial fission [25] and microtubule dynamics [24,26]. Loss of βIII-tubulin expression or its C-terminal tail may therefore provoke the dissociation of mitochondria from the microtubule cytoskeleton, thereby inducing slow mitochondrial dynamics together with increased mitochondrial fusion and clearance. Whether βIII-tubulin regulates mitochondrial dynamics by acting as a scaffold for mitochondrial–microtubule interactions or through its effects on microtubule dynamics remains to be clarified. In addition, recent studies have revealed that post-translational modifications on specific microtubule subsets regulates the localization of the endoplasmic reticulum, with flow-on effects on autophagic vesicle and lysosomal localization [20]. Further studies investigating if the effect of βIII-tubulin expression is specific to mitochondria, can be generalized more broadly to other organelles, or results from changes in endoplasmic reticulum localization, are warranted.

Mitochondrial rounding and swelling, fragmentation of the mitochondrial network, and a more centralized mitochondrial distribution observed in cells with suppressed βIII-tubulin expression is consistent with depolarization of the mitochondrial membrane potential and retrograde transport of mitochondria targeted for degradation [16,51,52,53,54]. While we did not identify any significant effect of βIII-tubulin expression on basal mitochondrial membrane potential or mitochondrial ultrastructure, it remains possible that the mitochondrial network architecture associated with suppressed βIII-tubulin expression may reflect dysregulated mitochondrial quality control and turnover. Altered mitochondrial quality control and decreased mitochondrial function may explain the loss of metabolic flexibility and increased reliance on glycolytic metabolism to support bioenergetic and anabolic demand when βIII-tubulin expression is suppressed [4].

A role for βIII-tubulin in modulating mitochondrial dynamics has profound implications on health and disease. High expression of βIII-tubulin is correlated with poor patient outcome, disease progression and chemotherapy resistance in a number of cancers, including non-small cell lung cancer (NSCLC) [4,44,48,57]. Mitochondrial dynamics are also critical in NSCLC tumor maintenance, and mitochondrial localization and trafficking can have a significant influence on cell function, including hypoxic signaling [58]. Since mitochondrial clearance promotes tumor progression in NSCLC [59,60], βIII-tubulin’s role in maintaining a functional active mitochondrial network may explain its association with aggressive advanced disease. Furthermore, recent studies highlighting the importance of localized metabolic control in driving focal adhesion turnover and cancer cell migration and invasion [61] suggest that the importance of βIII-tubulin in regulating these processes may support the metastasis of cancer cells that aberrantly express this protein. Certainly in neuronal cells, an extended mitochondrial network supports the metabolic requirements of long-range vesicular and organelle trafficking necessary to maintain cellular function across the entire neuronal cytoplasm [62].

While it is not yet clear if βIII-tubulin acts directly or indirectly on mitochondria, our analysis of protein–protein interactions has identified putative protein candidates that have the potential to directly mediate effects on mitochondrial architecture. PARK7 was identified as specifically interacting with βIII-tubulin and is known to play a role in mitochondrial quality control, mitochondrial trafficking, and in maintaining mitochondrial-ER network homeostasis [55,63]. PARK7 is localized mainly to the cytoplasm. In response to oxidative stress, it relocalizes to the cytoplasmic side of the mitochondria [56]. While further validation of this interaction to define its context dependency is required, whether oxidative stress regulates the association of this protein with βIII-tubulin is not yet clear. The role of PARK7/DJ-1 in regulating microtubule dynamics and ER stress in neurons [64], roles also played by βIII-tubulin [2,4], further supports their potential role in functionally coupling mitochondrial dynamics to βIII-tubulin in the microtubule cytoskeleton. Interestingly, suppression of PARK7 expression in a neuroblastoma-based neuronal model also suppressed βIII-tubulin expression and microtubule dynamics, supporting the functional coupling of βIII-tubulin and PARK7 [64]. PARK7 interacts with VDAC [55], which has been strongly implicated in tubulin–mitochondrial interactions and the regulation of mitochondrial metabolism with β-tubulin isotype specificity in cell-free systems [7,33,65]. The disruption of ER-mitochondrial contacts by disruption of PARK7-VDAC interactions at the mitochondrial membrane is thought to affect autophagosome formation and mitochondrial fission and fusion [55], supporting a role for βIII-tubulin in autophagosome, ER and mitochondrial dynamics demonstrated in previous studies [4], and implicated by the findings of this study. These mechanisms are also thought to play an important role in the progression of Parkinson’s Disease [55], suggesting that by regulating mitochondrial architecture, βIII-tubulin may impact diverse cellular functions with implications in neuronal biology as well as cancer and Parkinson’s Disease. These findings warrant further investigation to define the precise mechanisms by which βIII-tubulin regulates mitochondrial and ER homeostasis to impact stress response signaling.

Other minor β-tubulin isotypes (βII-tubulin and βIVα-tubulin) have also been associated with regulating mitochondrial localization specifically in cardiomyocytes [66,67,68,69], although the mechanisms by which they operate remain unknown. With increasing recognition for the combinatorial manner in which β-tubulin isotypes collectively regulate microtubule functions and microtubule-dependent processes in cells [18,21,70], it is likely that the tubulin code regulating mitochondrial dynamics is a complex spatiotemporal network that depends on the local β-tubulin composition.

This study provides the first evidence that modulating the expression of βIII-tubulin or its structural domains alters mitochondrial localization and architecture in an isotype-specific manner. This insight into the tubulin code warrants further investigation to define the precise direct or indirect molecular mechanisms mediated by this specific β-tubulin isotype, with potentially far-reaching implications for many facets of biology in health and disease.

## Figures and Tables

**Figure 1 cells-11-00776-f001:**
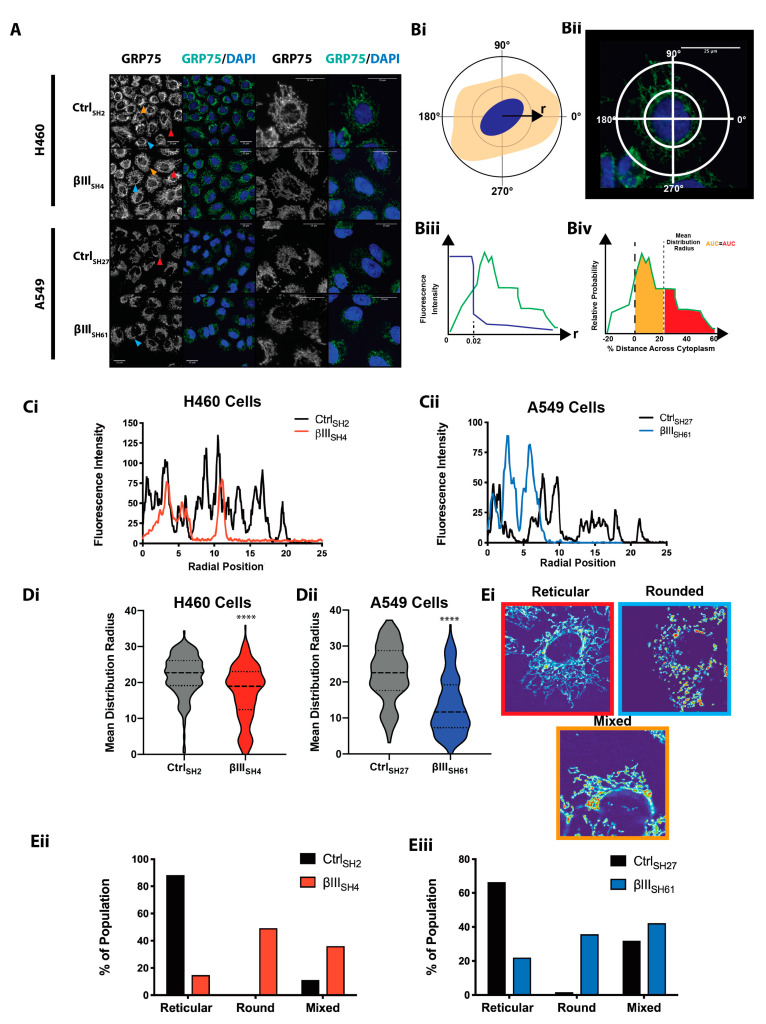
βIII-tubulin regulates mitochondrial network structure and morphology. (**A**) Mitochondrial network distribution by GRP75 immunofluorescence staining (left panel: greyscale; right panel: GRP75 green, DAPI blue) in NCI-H460 and A549 cells expressing control (NCI-H460 Ctrl_SH2_, A549 Ctrl_SH27_) and βIII-tubulin targeted shRNA (NCI-H460 βIII_SH4_, A549 βIII_SH61_). Red arrowhead: reticular mitochondrial morphology; blue arrowhead: rounded morphology; orange arrowhead: mixed morphology. Scale bars 25um. Representative of *n* = 3 independent experiments. Pseudocolored images show signal intensity from low (blue) to high (red) with high magnification images in Appendix A. (**B**) Workflow overview to measure the distribution of mitochondrial throughout the cytoplasm. Signal intensities for individual cells are transformed into polar co-ordinates defined as a radial distance from the centre of the nucleus at a specific angle (**Bi**,**Bii**). The edge of the nucleus and cytoplasm is defined (**Biii**) and the signal intensity is scaled to 0 to 100% corresponding to the centre of the nucleus (0%) to the edge of the cytoplasm (100%), with the edge of the nucleus at 25%. The mean distribution radius is defined at the radial position, r, where the integrated relative probability of the signal is equal from between the nucleus and r and between the edge of the cytoplasm and r (**Biv**). (**C**) Representative fluorescence intensity profile of GRP75 expression from the edge of the nucleus in NCI-H460 (**Ci**) and A549 (**Cii**) cells. (**D**) Mitochondrial mean distribution radius in NCI-H460 (**Di**) and A549 (**Dii**) cells. **** *p* < 0.0001. Data from the analysis of 30–60 cells from each condition for each of *n* = 3 independent experiments. (**Ei**) Representative images of reticular, rounded and mixed mitochondrial morphologies. Signal intensity is pseudocolored as a thermal look up table to convey signal distribution. The proportion of cells with rounded, reticular or mixed mitochondrial morphology in NCI-H460 (**Eii**) and A549 (**Eiii**) cells. Data from 100 cells from three independent experiments.

**Figure 2 cells-11-00776-f002:**
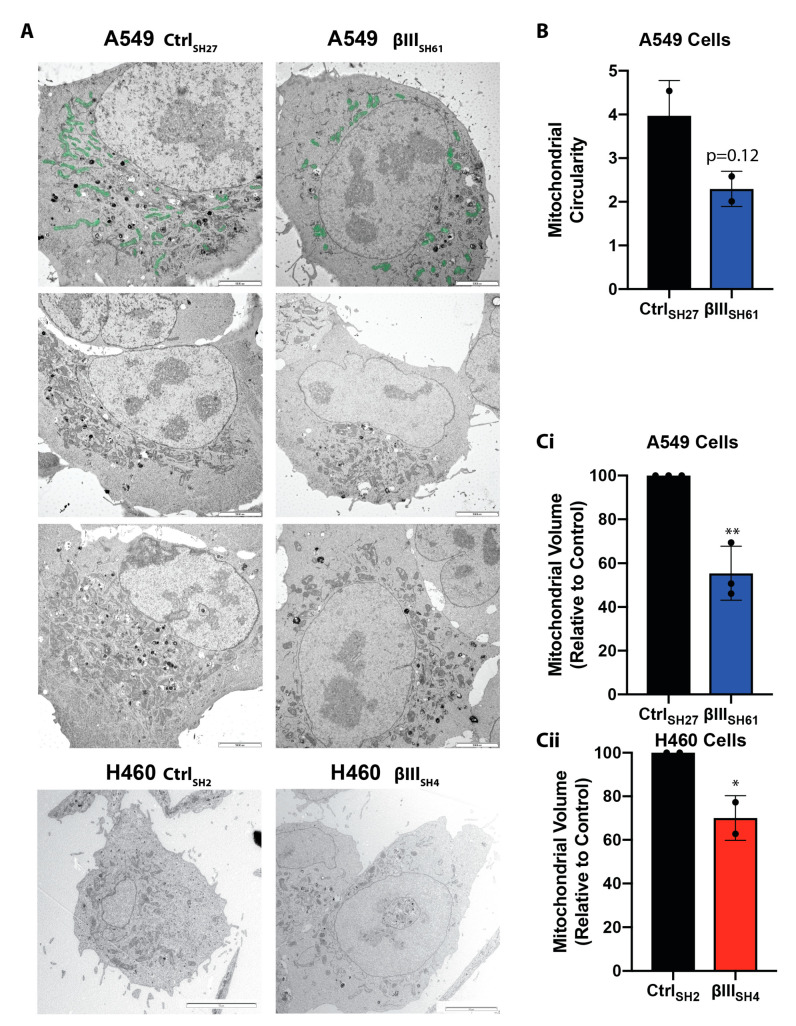
βIII-tubulin regulates mitochondrial volume. (**A**) Representative electron micrographs of NCI-H460 and A549 cells expressing control (H460 Ctrl_SH2_, A549 Ctrl_SH27_) and βIII-tubulin targeted shRNA (NCI-H460 βIII_SH4_, A549 βIII_SH61_). Mitochondria in the upper panels have been pseudocolored green to allow simple comparison of the two cell types (see Appendix A for uncolored original images). Scale bar 5 μm. (**B**) Mitochondrial circularity in A549 cells expressing control (A549 Ctrl_SH27_) and βIII-tubulin targeted shRNA (A549 βIII_SH61_) (mean ± SD of *n* = 2 independent experiments, *p* = 0.12). Points are the average of each independent experiment. (**C**) Mitochondrial volume in control and βIII-tubulin knockdown cells in A549 cells (**Ci**, mean ± SD of *n* = 3 independent experiments, *p* = 0.0033) and NCI-H460 cells (**Cii**, mean ± SD of *n* = 2 independent experiments, *p* = 0.0459). Points are the average of each independent experiment. * *p* < 0.05, ** *p* < 0.01.

**Figure 3 cells-11-00776-f003:**
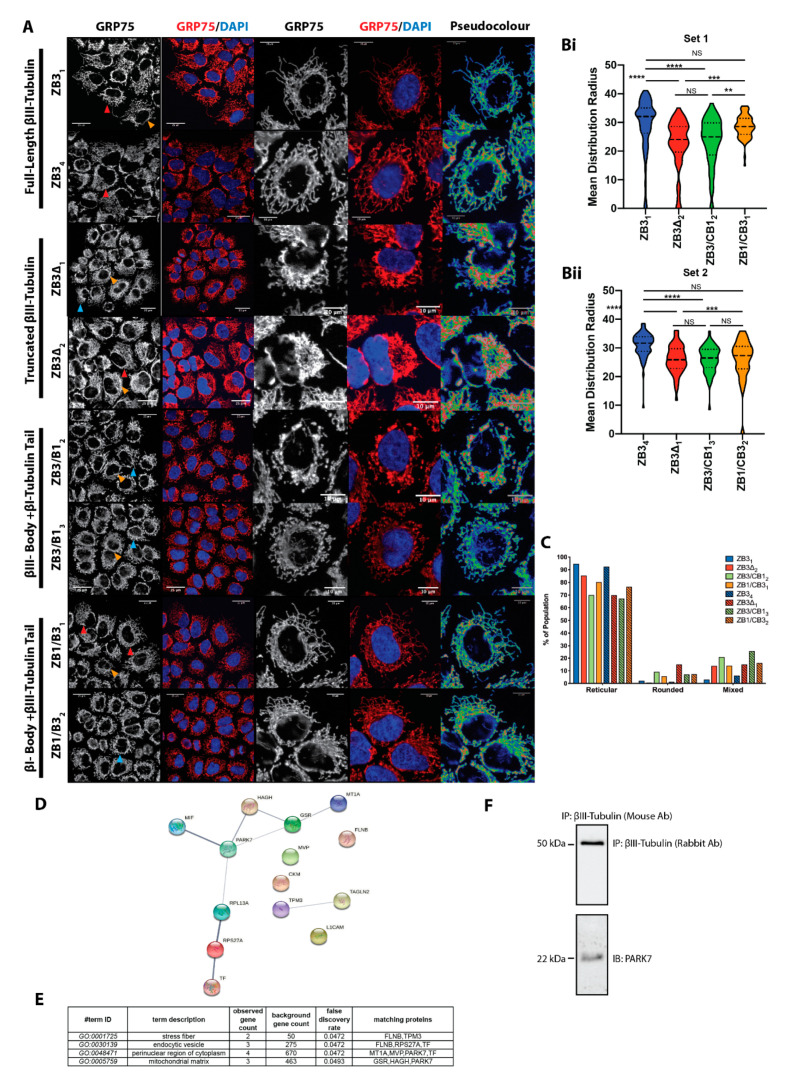
The βIII-tubulin C-terminal tail regulates mitochondrial network structure and morphology. (**A**) Mitochondrial network distribution by GRP75 immunostaining (left panel: greyscale; right panel: GRP75 green, DAPI blue) in NCI-H460 expressing the full-length βIII-tubulin protein (ZB3), truncated βIII-tubulin (ZB3Δ), βIII-tubulin with βI-tubulin C-terminal tail (ZB3/CB1), and βI-tubulin with βIII-tubulin C-terminal tail (ZB1/CB3). Scale bars 25 mm for lower magnification and 10 mm for higher magnification. Red arrowhead: reticular mitochondrial morphology; blue arrowhead: rounded morphology; orange arrowhead: mixed morphology. Scale bars 25mm. Pseudocolored images show signal intensity from low (blue) to high (red). Representative of *n* = 3 independent experiments. (**B**) Mitochondrial mean distribution radius for Set 1 (**Bi**) and Set 2 (**Bii**) gene-edited cells. Non-parametric ANOVA with Dunn’s correction for multiple comparisons. NS: non-significant, ** *p* < 0.01, *** *p* < 0.001, **** *p* < 0.0001. Data from the analysis of more than 45 cells from *n* = 3 independent experiments. (**C**) The proportion of cells with rounded, reticular or mixed mitochondrial morphology. Data from at least 120 cells per clone from three independent experiments. (**D**) Interaction network of proteins that exclusively interact with βIII-tubulin. (**E**) Functional annotation analysis identifies enrichment of βIII-tubulin-associated proteins in cellular compartments (GO terms). (**F**) Validation of the interaction between βIII-tubulin and PARK7 by western blotting (representative of *n* = 3 independent experiments).

**Figure 4 cells-11-00776-f004:**
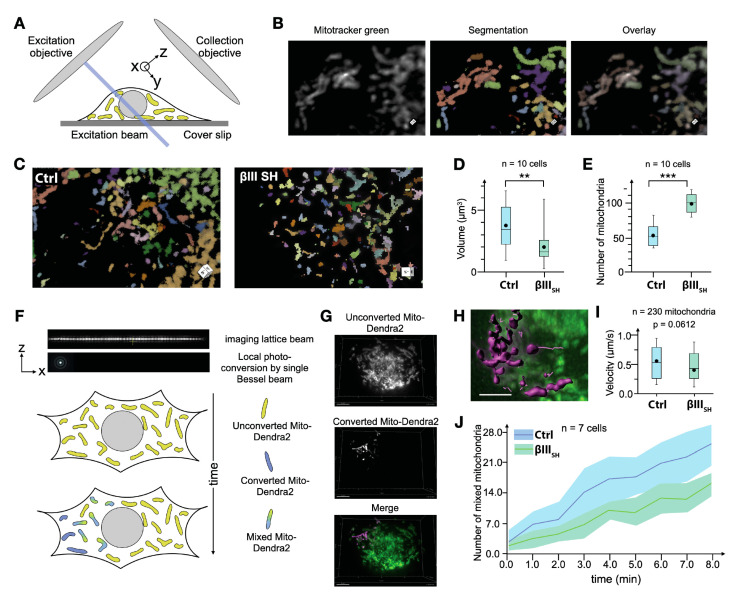
βIII-tubulin regulates mitochondrial dynamics. (**A**) Schematic of mitochondria imaging using the lattice light-sheet microscope. The blue line represents the plane of excitation. (**B**) Representative image of MitoTracker Green intensity, segmentation and overlay. (**C**) Mitochondria in control cells have larger volume and show high-aspect ratio in comparison to cells expressing βIII-tubulin targeted shRNA (βIII_SH_). (**D**) Mitochondrial volumes as measured from segmentation of 3D images.** *p* < 0.01. (**E**) Total number of mitochondria in whole cells, *** *p* < 0.001. (**F**) Schematic of single Bessel beam-based photo-conversion of Mito-Dendra2 to track individual mitochondria particles and fusions. Axes shown correspond to the geometry depicted in (**A**). (**G**) A representative image of a cell showing mito-Dendra2 post-photoconversion. (**H**) Segmentation and tracking of mitochondria in the converted channel enable quantitation of mitochondrial fusion. (**I**) Mitochondrial speed in control cells and in cells expressing βIII-tubulin targeted shRNA (βIII_SH_), *p* = 0.0612 (**J**) Quantitation of double-colored mitochondria as a readout of mitochondrial fusion and mixing, *n* = 3 independent experiments. Ctrl = Control cells; βIII_SH_ = cells expressing βIII-tubulin targeted shRNA.

## Data Availability

Information and requests for Matlab codes should be directed to the Lead Contact Maria Kavallaris.

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
