# Peer review of "βIII-Tubulin Structural Domains Regulate Mitochondrial Network Architecture in an Isotype-Specific Manner"

_cells, 2022, doi:10.3390/cells11050776_

Round 1
Reviewer 1 Report
In this paper, Parker et al. showed that bIII-tubulin regulates the architecture and dynamics of the mitochondria network in an isotype-specific manner. They observed a centralized mitochondrial distribution, mitochondria rounding, and network fragmentation in bIII-tubulin knock-out cells. They also revealed that the C-terminal tail of bIII-tubulin contributes to regulating mitochondrial architecture.
Major comments
- In Figure 3F, the authors should add an image that shows PARK7 is not precipitated with bI-tubulin. They should conduct immunoprecipitation using the deletion mutant and chimera proteins to confirm PARK7 and other interacting proteins bind to the C-terminal tail of bIII-tubulin.
- The author showed the critical roles of bIII-tubulin on the distribution and morphology of mitochondria. They should also examine whether the lack of bIII-tubulin affects mitochondrial functions such as oxygen consumption, membrane potential, or ROS production.
Minor Comments
- Images of figures 1A and 3A are too small to assess the mitochondrial morphology.
- ‘Results and Discussion” on p.5 should be “Results.”
Author Response
The authors gratefully thank the reviewer for their critical review of this manuscript
examining the role of bIII-tubulin and its structural domains in mitochondrial network
organization and architecture. We believe that the changes implemented in response to the
reviewers’ comments have strengthened the manuscript.
Please find below our response addressing the reviewers’ comments.
Reviewer 1.
1) The authors should show that PARK7 is not precipitated with bI-tubulin. They should
conduct immunoprecipitation to confirm that PARK7 and other interacting proteins bind
to the C-terminal tail of b III-tubulin.
The authors thank the reviewer for this suggestion. We have performed immunoprecipitation
of βI-tubulin as well as bIII-tubulin and have identified that PARK7 is indeed enriched in the
betaIII-tubulin immunoprecipitation compared with the bI-tubulin immunoprecipitation. It is
important to note that, as is also the case in the mass spectrometry data and indicated on
lines 382-383, multiple beta-tubulin isotypes (including bI-tubulin and bIII-tubulin) were
identified in the immunoprecipitation eluate (e.g. faint bIII-tubulin band in the bI-tubulin IP
in Figure 1), albeit at very low concentrations. Importantly, the PARK7 in the
immunoprecipitation eluates correlate with the amount of bIII-tubulin also pulled down in
both the bI-tubulin and bIII-tubulin immunoprecipitations, supporting the observations from
mass spectrometry data that PARK7 preferentially interacts with bIII-tubulin.
The difficulty in isolating one b-tubulin isotype from other isotypes in immunoprecipitation
experiments reflects the high affinity of tubulin proteins with one another and the fact that
they interact collectively within the microtubule network as filaments (Janke and Magiera,
2020). As a result, immunoprecipitation conditions that are stringent enough to isolate one
tubulin isotype from other tubulin isotypes also interfere with the interaction of b-tubulins
with the non-tubulin proteins that they interact with, and therefore cannot be used to
address the research question posed in this study. Therefore, we sought to use conditions
that adequately enriched for the target beta-tubulin isotype as the predominant b-tubulin
isotype in the eluate.
An additional line has been added on line 384 to clarify that the immunoprecipitation
conditions used significantly enrich for betaIII over bI-tubulin and therefore proteins
identified in these eluates are likely to bind to bIII-tubulin.
Figure 1. b-Tubulin- and bIII-tubulin immunoprecipitation in H460 CtrlSH2 cells showing
enrichment of betaIII-tubulin in the bIII-tubulin immunoprecipitation as expected. Due to the
strong interaction between tubulin isotypes, betaIII-tubulin is also present in the bI-tubulin
immunoprecipitation, although at a very low concentration. The association with PARK7
mimics the pattern seen for bIII-tubulin in both immunoprecipitations, supporting an
interaction of this protein with bIII-tubulin.
As indicated in the manuscript on lines 372-397 and 541-568, we have indicated that our
findings of bIII-tubulin-interacting protein network does not discern between proteins that
interact with the body or tail of this tubulin isoform. Performing these immunoprecipitation
experiments on the gene-edited cells to dissect the contribution of bIII-tubulin structural
domains to these protein-protein interactions requires substantial development of new
methodology due to the unique C-terminal tail sequences of the modified bIII-tubulin
proteins. An examination of how these proteins interact with bIII-tubulin is the subject of a
follow-up manuscript that examines these protein-protein interactions in detail. As such, we
believe that a thorough analysis of how these proteins interact with bIII-tubulin structural
domains falls outside the scope of this short communication.
2) The authors should examine whether the lack of bIII-tubulin affects mitochondrial
functions such as oxygen consumption, membrane potential or ROS production.
The authors thank the reviewer for this suggestion and agree that the strong association of
mitochondrial architecture with mitochondrial function observed in other contexts (Gomes
et al., 2011; MacAskill and Kittler, 2010; Rambold et al., 2011; Rossignol et al., 2004) suggests
that our observations of changes in the mitochondrial architecture may be associated with
functional changes in the mitochondria, as indicated in lines 280-293.
As indicated in Supplementary Figure 1E and on lines 283-290, we had assessed mitochondrial
membrane potential differences in control and knock-down clones using JC-1 staining and
FACS and identified no significant difference in mitochondrial membrane potential. As
indicated on lines 107-113 and 539-542 in reference to our previous work (Parker et al., 2016),
our detailed analysis of mitochondrial metabolic function showed that suppression of bIIItubulin
expression did not affect the basal oxygen consumption rate or maximum respiratory
capacity of cells, but did increase the rate of glycolytic metabolism in basal conditions, and
suppressed the ability of cells to switch from glycolytic to oxidative phosphorylation in
glucose-depleted conditions (Parker et al., 2016). Our more recent analysis of mitochondrial
function in gene-edited cells expressing the modified tubulin proteins did not show any
difference in glycolytic or oxidative phosphorylation with expression of structurally modified
βIII-tubulin proteins (Figure 2).
Figure 2. Metabolic profile of cells expressing structurally modified bIII-tubulin proteins as
measured using the Seahorse XF Analyzer. A) Oxygen consumption rate. Mean SEM of n=6
independent experiments. No significant differences comparing cells expressing structurally
modified betaIII-tubulin proteins against cells expressing the full length bIII-tubulin protein
(ZB31, ZB34) for each respective set of gene-edited cells; ANOVA with Kruskal-Wallis
correction for multiple comparisons. H460 cells with control and bIII-tubulin knock down are
included as technical controls as indicated in (Parker et al., 2016); no significant difference;
Mann-Whitney U-test. B) Extracellular Acidification Rate. Mean SEM of n=6 independent
experiments. No significant differences comparing cells expressing structurally modified bIIItubulin
proteins against cells expressing the full length bIII-tubulin protein (ZB31, ZB34) for
each respective set of gene-edited cells except for ZB34 vs ZB1/CB32 (*p=0.041); ANOVA with
Kruskal-Wallis correction for multiple comparisons. H460 cells with control (CtrlSH2) and
betaIII-tubulin knock down (bIIlSH4) are included as technical controls as indicated in (Parker
et al., 2016); *p<0.05; Mann-Whitney U-test for comparison of CtrlSH2 and bIIlSH4.
Minor Comments
1) Images of figures 1A and 3A are too small to assess the mitochondrial morphology.
The sizes of Figures 1A and 3A have been increased in the revised manuscript. Additional
representative images pseudocolored by signal intensity have been added to both Figures 1A
and 3A to more clearly indicate the distribution of the mitochondrial signal. In addition, high
resolution images have also been included in Supplementary Figure 1A to enable assessment
of mitochondrial morphology.
2) Results and Discussion should be “Results”.
The authors thank the reviewer for alerting us to this error. The title for this section has been
corrected to “Results”.
References for Reviewer 1 and 2
Gan, P.P., McCarroll, J.A., Po’uha, S.T., Kamath, K., Jordan, M.A., and Kavallaris, M. (2010).
Microtubule dynamics, mitotic arrest, and apoptosis: drug-induced differential effects of
betaIII-tubulin. Mol. Cancer Ther. 9, 1339–1348.
Gardner, L.B., Li, F., Yang, X., and Dang, C.V. (2003). Anoxic Fibroblasts Activate a Replication
Checkpoint That Is Bypassed By E1a. Mol. Cell. Biol. 23, 9032–9045.
Gomes, L.C., Di Benedetto, G., and Scorrano, L. (2011). During autophagy mitochondria
elongate, are spared from degradation and sustain cell viability. Nat. Cell Biol. 13, 589–598.
Janke, C., and Magiera, M.M. (2020). The tubulin code and its role in controlling microtubule
properties and functions. Nat. Rev. Mol. Cell Biol. 21, 307–326.
Kumar, S., and Vaidya, M. (2016). Hypoxia inhibits mesenchymal stem cell proliferation
through HIF1α-dependent regulation of P27. Mol. Cell. Biochem. 415, 29–38.
MacAskill, A.F., and Kittler, J.T. (2010). Control of mitochondrial transport and localization in
neurons. Trends Cell Biol. 20, 102–112.
McCarroll, J.A., Gan, P.P., Liu, M., and Kavallaris, M. (2010). betaIII-tubulin is a multifunctional
protein involved in drug sensitivity and tumorigenesis in non-small cell lung cancer. Cancer
Res. 70, 4995–5003.
McCarroll, J.A., Gan, P.P., Erlich, R.B., Liu, M., Dwarte, T., Sagnella, S.S., Akerfeldt, M.C., Yang,
L., Parker, A.L., Chang, M.H., et al. (2015). TUBB3/βIII-tubulin acts through the PTEN/AKT
signaling axis to promote tumorigenesis and anoikis resistance in non-small cell lung cancer.
Cancer Res. 75, 415–425.
Parker, A.L., Turner, N., McCarroll, J.A., and Kavallaris, M. (2016). βIII-Tubulin alters glucose
metabolism and stress response signaling to promote cell survival and proliferation in
glucose-starved non-small cell lung cancer cells. Carcinogenesis 37, 787–798.
Parker, A.L., Teo, W.S., Pandzic, E., Vicente, J.J., McCarroll, J.A., Wordeman, L., and Kavallaris,
M. (2018). β-tubulin carboxy-terminal tails exhibit isotype-specific effects on microtubule
dynamics in human gene-edited cells. Life Sci. Alliance 1.
Rambold, A.S., Kostelecky, B., Elia, N., and Lippincott-Schwartz, J. (2011). Tubular network
formation protects mitochondria from autophagosomal degradation during nutrient
starvation. Proc. Natl. Acad. Sci. USA 108, 10190–10195.
Rossignol, R., Gilkerson, R., Aggeler, R., Yamagata, K., Remington, S.J., and Capaldi, R.A.
(2004). Energy substrate modulates mitochondrial structure and oxidative capacity in cancer
cells. Cancer Res. 64, 985–993.
Teramura, T., Onodera, Y., Takehara, T., Frampton, J., Matsuoka, T., Ito, S., Nakagawa, K., Miki,
Y., Hosoi, Y., Hamanishi, C., et al. (2013). Induction of functional mesenchymal stem cells from
rabbit embryonic stem cells by exposure to severe hypoxic conditions. Cell Transplant. 22,
309–329.

Reviewer 2 Report
I have read with interest the manuscript by Parker et al. I have a good impression of all the work done and for me (although I am more an expert in biochemistry and structural biology) the manuscript seems sonvincing.
I have a suggestion that I think can improve the manuscript for not so specialized readers.
Some cell cycle phases consume more oxygen than others, thus, under hypoxic or anoxic conditions the distribution of cells through the cell cycle phases changes when compared with normal (normoxic) conditions and cells accumulate in the phases of more ATP requirements (S phase, anaphase). I assume that the observed effects on reduction of mitocondrial volume and the changes in mitochondrial network morphology in the absence of the BIII C-terminal tail, would have an effect on cell cycle distribution similar to mild or strong hypoxia. Could the authors check by FACS and propidium iodide analysis if this is the case? Have they observed mitotic aberrations?. I would like to have this experiment (not too complex) done before publications.
https://www.researchgate.net/publication/8984330_Anoxic_Fibroblasts_Activate_a_Replication_Checkpoint_That_Is_Bypassed_By_E1a
https://www.researchgate.net/publication/296193758_Hypoxia_inhibits_mesenchymal_stem_cell_proliferation_through_HIF1a-dependent_regulation_of_P27
https://www.researchgate.net/publication/230789689_Induction_of_Functional_Mesenchymal_Stem_Cells_From_Rabbit_Embryonic_Stem_Cells_by_Exposure_to_Severe_Hypoxic_Conditions
Author Response
Refer to attachment for point by point response

Round 2
Reviewer 1 Report
The revised manuscript is acceptable.
Reviewer 2 Report
The authors have addressed my concerns.